# Effectiveness of an Educational Filmmaking Project in Promoting the Psychological Well-Being of Adolescents with Emotive/Behavioural Problems

**DOI:** 10.3390/healthcare11121695

**Published:** 2023-06-09

**Authors:** Antonella Gagliano, Carola Costanza, Marzia Bazzoni, Ludovica Falcioni, Micaela Rizzi, Costanza Scaffidi Abbate, Luigi Vetri, Michele Roccella, Massimo Guglielmi, Filippo Livio, Massimo Ingrassia, Loredana Benedetto

**Affiliations:** 1Department of Human and Pediatric Pathology “Gaetano Barresi”, University of Messina, 98122 Messina, Italy; antonellagagliano.npi@gmail.com (A.G.);; 2Department of Sciences for Health Promotion and Mother and Child Care “G. D’Alessandro”, University of Palermo, 90128 Palermo, Italy; carola.costanza@unipa.it; 3University of Cagliari & “A. Cao” Pediatric Hospital, Brotzu Hospital Trust, 09047 Cagliari, Italy; 4Department of Psychology, Educational Science and Human Movement, University of Palermo, 90141 Palermo, Italy; michele.roccella@unipa.it; 5Oasi Research Institute-IRCCS, Via Conte Ruggero 73, 94018 Troina, Italy; 6Rehabilitation and Education Center “Dismed Onlus-Centro Studi per Le Disabilita’ del Mediterraneo”, 98100 Messina, Italy; 7Department of Clinical and Experimental Medicine, University of Messina, 98122 Messina, Italy

**Keywords:** mental health care, adolescence, internalizing/externalizing problems, social skills, neurodevelopmental disorders, filmmaking intervention

## Abstract

Evidence suggests that adolescents respond positively to simple, early interventions, including psychosocial support and educational interventions, even when offered in non-clinical settings. Cinematherapy can help manage life challenges, develop new skills, increase awareness, and offer new ways of thinking about specific problems. This pilot trial was conducted in Italy, aiming to investigate the effects of a six-week filmmaking course on the psychological well-being of adolescents (N = 52) with emotional/behavioural problems and neurodevelopmental disorders. At the end of the project, most participants showed improvements mostly in social skills, such as social cognition (*p* = 0.049), communication (*p* = 0.009), and motivation (*p* = 0.03), detected using the SRS Social Responsiveness Scale. In addition, social awareness (*p* = 0.001) increased in all patients. Statistically significant differences resulted in four sub-scales of Youth Self-Report Scale: withdrawn/depressed (*p* = 0.007), social problems (*p* = 0.003), thought problems (*p* < 0.001), and rule-breaking behaviour (*p* = 0.03); these results showed a decrease in emotional and behavioural problems. This study is an innovative therapeutic and educational approach based on the filmmaking art. This research can offer an empirical basis for the effectiveness of alternative therapeutic tools in child and adolescent psychiatric disorders. At the same time, it can be replicated in broader contexts (e.g., school and communities) to promote children’s psychological well-being.

## 1. Introduction

Over the past 30 years, a large increase in the number of children and adolescents suffering from anxiety and major depressive disorders or behavioural and conduct disorders has been described. Worldwide, 8.8% of the paediatric population has been diagnosed with several mental illnesses, accounting for a heavy disease burden [1]. Currently, it has become a real psychiatric emergency, exacerbated by the pandemic period, that leads many children to go through moments of sadness and emotional distress. The effects of the three years of the COVID-19 pandemic and its health security measures have had a strong emotional impact on people’s lives worldwide [2]. Among the most important effects are the psychological and social ones, which are insidious, as they condition the mental health of children and adolescents, with the risk of continuing into their future. Many of them have experienced episodes or states of depression and anxiety due, for example, to school closures and the consequent social isolation and physical restrictions [3]. Currently, between 30 and 40 per cent of the school-age population suffer from a psychiatric condition [4,5].

This represents a challenge for the mental health care system, with a growing need for new therapeutic and preventive interventions, such as those based on music, theatre, and film therapy. It is urgent to focus not only on diagnostic and therapeutic procedures but also on creating opportunities for personal growth and empowering positive experiences with people dealing with the same obstacles. Evidence suggests that adolescents respond positively to early interventions, including psychosocial support and educational interventions, when offered in non-clinical settings [6]. 

Psychosocial interventions include art and creative therapies. A small body of literature supports using art therapy to empower youth [6]. Regarding young people, participation in the arts is thought to increase confidence and self-esteem [7]. It has also been argued extensively in the literature that models of youth participation benefit mental health over the longer term [8]. As reported by previous studies [9], the inclusion of arts in mental health programs can assist people in developing and maintaining a positive identity and creating a sense of meaning and purpose in their lives. As a matter of fact, an improvement in self-esteem and social awareness was detected in our work.

Although video-based intervention and filmmaking have been previously studied in adult psychotherapy [10], their efficacy with children and adolescents still needs to be explored. In fact, in the current literature, very few studies on the recovery of social skills for children with autism spectrum disorder or intellectual disabilities are available [11,12]. Studies documenting the usefulness of filmmaking programs specifically designed for children and adolescents with neurodevelopmental and emotional disorders are very rare. Some film-based interventions have been proposed to adolescents as educational tools, but their efficacy in mental health education remains largely under-explored [13].

Differently from other figurative arts, cinema allows for different psychological benefits, such as the identification with the character’s feelings and emotions, the cathartic effect (learning through the character’s experiences), and the improvement in social abilities through sharing emotions and experiences with other participants. There is clinical evidence that through the use of movies and follow-up sessions, pre-adolescent children whose parents were going through a divorce improved their capacity to identify and express their feelings and developed coping skills [14,15]. A monthly cinema therapy group session, proposed to a group of girls, allowed them to identify and work through anger and conflicts and to mark their feelings about their families and social relationships. Starting from this evidence, we argue that the cooperative work required for making a film is a collective experience which may have a stronger impact on both emotions and thoughts than the vision of a film made from other people.

Creativity should be an essential element of the education process of people with disabilities and mental disorders [12]. Indeed, working on a film project could be an opportunity to create stories and characters and identify with them while maintaining distance. In this way, individuals are exposed to their psycho-physical difficulties through the stories of the characters on the screen who are coping with the same issues as the patients [16]. Such an approach is also a strong stimulus to consider their problems from a different perspective. Consequently, this can help people to ponder their own life, promoting self-exploration and improving their ability to externalize problems [17]. 

This study shows how a film-based intervention, with typical structural and practical phases of the filmmaking process, can actively involve children and adolescents with mental health problems. This study has several objectives. 

The first objective was therapeutic and psycho-educational. It intended to represent an educational and growth experience for children and adolescents with neurodevelopmental/psychiatric disorders to make them capable of putting their skills to the test by experimenting with the various roles that cinema caters to (screenplay, direction, set design, costumes, acting, video shooting, editing). At the same time, it wanted to stimulate them to live out highly creative and cooperative work experiences. In this way, the higher-order cognitive functions (executive functions) and the energetic, motivational, and empathic aspects (warm executive functions) have been strongly stimulated and trained. The constant presence of child neuropsychiatrists and psychologists, who supported the film director, ensured the achievement of these objectives. 

The second objective was to offer a theoretical and practical introduction to the key principles of filmmaking. The participants attended a series of structured lessons on the skills necessary to use the cinematographic language (from brainstorming and storyboarding to camera use or video editing) with the purpose of creating a professional audiovisual product. All participants and professionals were members of the technical and artistic cast, working under the guidance of a qualified movie director and cinema teacher. The students also received a training certificate. 

Finally, the current study explores whether, and to what extent, involvement in a filmmaking program could promote the psychological well-being of children and adolescents with emotive/behavioural problems. The hypothesis is that this highly creative and cooperative group experience could both enhance participants’ self-esteem and reduce the severity of their emotional/behavioural problems.

Therefore, the research project included a quantitative assessment through standardized measures of the participants’ emotional, social, and behavioural profiles. In addition, the benefits of the filmmaking course have been estimated on the extent of changes in the study measures (self-esteem and emotional/behavioural problems) through a before–after comparison. Furthermore, to support the outcomes of the filmmaking intervention, the evaluation of improvements observed by the caregivers was implemented.

## 2. Materials and Methods

### 2.1. Participants

The study involved 52 participants aged between 9 and 17 years: 24 girls (mean age 114 Mgirls = 16, SDgirls = 2.4) and 28 boys (age Mboys = 14.8, SDboys = 2.6), all with a diagnosis of neurodevelopmental/psychiatric conditions. The project was developed in two consecutive courses of 6 weeks each. A first group of participants (N = 18) was selected among patients from an outpatient service at the Child and Adolescent Neuropsychiatry Unit, “A. Cao” Hospital—“G. Brotzu” Hospital Trust, Cagliari. A second group (n = 34) was chosen at Rehabilitation and Education Center “Dismed Onlus-Centro Studi per Le Disabilita’ del Mediterraneo”, Messina, Italy.

Inclusion criteria were: (i) clinical diagnosis of neurodevelopmental/psychiatric conditions, (ii) aged under 17 years old.

Information regarding personal data (i.e., gender and date of birth), personal history (i.e., previous diagnosis), and family history (psychiatric conditions) was collected.

The several diagnoses in the total sample were, in descending order: anxiety disorder (n = 19), ADHD (n = 15), depressive disorder (n = 14), dyslexia (n = 13), adjustment disorder or PTSD (n = 12), autism spectrum disorder (n = 11), OCD (n = 7), bipolar disorder (n = 6), psychotic disorder (n = 5), Tourette syndrome (n = 5). Most of the enrolled adolescents received more than one diagnosis. The exclusion criteria were Total IQ (TIQ) < 70, assessed using the Wechsler scale (WISC-IV). 

### 2.2. Aims of the Study 

This study explores if a therapeutic approach based on a six-week filmmaking course would help adolescents with emotional/behavioural problems and neurodevelopmental disorders. We hypothesized that an intensive training course aimed to actualize an artistic and creative product, such as a film, could lead to improving emotional and social abilities of a group of adolescents. We questioned if the peer interaction and the cooperative work on a film set could represent an effective tool for promoting adolescents’ psychological well-being, as reported in most of the scientific literature [13,18]. 

### 2.3. Procedure and Phases

Different professionals (psychologists, child psychiatrists, educators) supervised all the project phases to provide appropriate support during the entire project. 

An experienced movie director [19] coordinated the whole process, with the help of a professional actor who supported the participants as an acting coach [20]. 

First, the invitation to participate in the project was proposed to parents through the two clinical/rehabilitation centres. The interested parents were invited for a meeting during which they obtained all details and scope of the project and provided written consent for their child’s participation. Subsequently, children participated in a meeting to receive information about the filmmaking project and to obtain their assent for voluntary involvement.

Then, participants started a daily course of 5–6 h per day, lasting in total three weeks. During the first two days, all project members attended theory-based lessons in filmmaking history and techniques. This phase also focused on building trust relationships among the participants. 

From the third day onward, the three main stages of the film production process were structured. First of all, a pre-production workshop took place. In this phase, the participants, divided into groups, ideated and wrote many screenplays. Participants were assigned different filmmaking roles according to their interests and attitudes: scriptwriting, direction, photography, sound, costume design, make-up artistry, production organization, and digital editing. Some subjects also held more than one role. During the pre-production workshop, the participants were also engaged in storyboarding, spaces were selected, and equipment for sets was provided. A day-by-day film production timeline was also created. Additionally, following the creative (i.e., choice of the topic, narratives, roles, etc.) and technical phases (i.e., training on camera functioning), the filmmaking process took place in natural locations or public areas (sports ground, squares, etc.) that were chosen by the participants. Participants were expected to produce a few short films (lasting about 15 min), but they outperformed expectations and ultimately produced 13 short films in total (6 in the first edition and 7 in the second edition of the course). The script themes were largely inspired by the main adolescents’ issues and concerns, such as unintentional injuries, violence, mental health, suicide, self-cutting, alcohol and drug use, gender dysphoria, sense of unease, etc. In addition, some scripts were inspired by social and political issues, such as the Russia–Ukraine war and the fear of the apocalypse. Only a few themes contain hopeful messages that aim to promote the power of friendship and love. However, every script and film told a good story with an exciting plot that promoted reflection on sensitive topics. 

Finally, during the last two days of the programme, the video-editing crew worked with the movie director for the post-production phase: firstly, they drafted a rough cut of the film; then, they began to review and edit the footage making additions such as visual effects, background music, and sound design.

### 2.4. Questionnaires

Both the participants and their caregivers were asked to complete the following self-administered questionnaires at the beginning (T_0_) and at the end (T_1_) of the course (Appendix A): 

Rosenberg Self Esteem Scale (RSE [21,22]); Youth Self-Report scale (YSR; [23,24], Italian adaption by Frigerio A. et al., 2004 [25]); Social Responsiveness Scale (SRS; [26], Italian adaption Zuddas et al., 2010 [27]); and Revised Conners’ Parent Rating Scale (CPRS-R; Conners, 1997, [28] Italian adaption Nobile, Alberti & Zuddas, 2007 [29]).

Firstly, children and adolescents completed the Rosenberg Self Esteem Scale, which is a 10-item questionnaire that assesses global self-esteem. All items have a 4-point Likert scale (ranging from “strongly disagree” to “strongly agree”, half reversed). Higher scores indicate a higher sense of self-esteem and scores below 15 suggest low self-esteem. The Italian questionnaire has shown good internal validity (Cronbach’s α = 0.84) and test–retest reliability (r = 0.76; Prezza et al., 1994 [22]).

Secondly, they filled out the Youth Self-Report Scale, which measures the severity of emotional and behavioural problems based on a sense of self-estimated evaluation. The YSR is recognized as having good psychometric characteristics, with α values ranging from 0.71 to 0.95. The questionnaire (112 items) is an instrument of the ASEBA (Achenbach System of Empirically Based Assessment; Achenbach, 1993 [23]) and includes 8 sub-scales or syndromes: anxious/depressed, withdrawn/depressed, somatic complaints, social problems, thought problems, attention problems, rule-breaking behaviour, and aggressive behaviour. Patients select responses (not true/somewhat or sometimes true/very true or often true), and syndrome scores are then calculated by summing the response of each problem item (e.g., “I am too shy or timid” and “I threaten to hurt people”). Higher scores indicate more severe levels of the measured syndromes.

Finally, all primary family caregivers were asked to complete Social Responsiveness Scale (SRS) and the Revised Conners’ Parent Rating Scale (CPRS-R) at points T_0_ and T_1_. 

The Social Responsiveness Scale (SRS) measures the severity of social impairments in children affected by autism spectrum disorder or children from the general population. This questionnaire has excellent psychometric characteristics, and it has also been proven to be a useful tool for verifying changes in the severity of social impairment following intervention [30]. 

The 65-item questionnaire includes five subscales: social awareness, social cognition, social communication, social motivation, and autistic mannerisms. Caregivers respond by estimating to what extent the item describes a child’s observed behaviour on a 4-point Likert scale (from “not true” to “almost always true”, “Seems much more nervous in social situations than when alone”). Higher scores indicate higher severity of social deficits in that area. 

The Revised Conners’ Parent Rating Scale (CPRS-R; Conners, 1997, [28] Italian adaption Nobile, Alberti & Zuddas, 2007 [29]) is a questionnaire measuring parental perception of a child’s behavioural problems. The short form (27 items) comprises four subscales: oppositional behaviour, cognitive problems, hyperactivity, and ADHD index. The respondent rates on a 4-point scale how often the child engages in the described behaviour (from “not true at all/never, seldom” to “very much true/very often, very frequent”; for example, “Inattentive, easily distracted”). Higher scores indicate more severe behavioural problems than lower ones. This short form of the CPRS-R proved to be a valid tool and sensitive as a treatment outcome measure [31].

## 3. Results

Statistical procedures were performed using IBM SPSS 19.0 data analysis software. All participants recruited at T_0_ completed the filmmaking course with no dropouts, so the measurements (n = 52) at the two assessment points (T_0_ vs. T_1_) from both children and caregivers were available for the statistical analysis.

Firstly, descriptive statistics for dependent variables were calculated (see Table 1). The assumption of normality for all measures was verified by the Kolmogorov–Smirnov, with Lilliefors correction and Shapiro–Wilk tests: all *p*
*<* 0.05. Therefore, Wilcoxon signed-rank tests were performed to estimate pairwise differences between the measures. These tests were employed as a non-parametric alternative to the paired *t*-test, as it does not rely on the assumption of normality. Specifically, they were utilized to identify significant differences (*p* < 0.05) between the YSR, SRS, and CPRS-R measures (T scores) at two different time points (T_0_ vs. T_1_). Results are reported in Table 2.

Primarily, an increase in self-esteem scores (RSE) can be observed (see Table 1), even though the differences between phases do not reach statistical significance. Nevertheless, the number of participants at T_0_ with low vs. normal/high self-esteem was 16 (30.8%) and 36 (69.2%), respectively; at T_1_, the percentage of participants with low self-esteem was halved (8 participants, 15.4%), and 44 participants (84.6%) reported normal/high self-esteem levels. Then, the number/percentages of participants with high vs. low RSE were tested for the independence of distributions at T_0_ and T_1_ using the chi-square test (χ^2^). However, the test showed no significant association between self-esteem levels and phases: χ^2^ (1) = 1.56, *p* = 0.21.

Moreover, all the subscales measured by the Youth Self-Report scale were decreased from T_0_ to T_1_. A significant decrease in YSR scores from T_0_ to T_1_ resulted in four sub-scales: withdrawn/depressed (*p* = 0.005), social problems (*p* = 0.007), thought problems (*p* < 0.001), and rule-breaking behaviour (*p* = 0.04) (see Table 2).

Moreover, in the Social Responsiveness Scale (see Table 2), all parameters decreased significantly between T_0_ and T_1_, with the only exception being autistic mannerisms. Most importantly, the parameters that reached the widest statistically significant improvement were the social awareness subscale (*p* = 0.001) and social motivation subscale (*p* = 0.03). Social awareness increased in all patients. 

Finally, while a decrease in CPRS-R scores was observed (subscales and ADHD index), the differences (T_0_ vs. T_1_) did not reach statistical significance. This finding was highly predictable since the programme was too short to change core aspects of a disorder, such as hyperactivity or oppositional behaviour. 

## 4. Narrative and Qualitative Reports

In addition to the numerical data that we have collected and analysed via statistical methods, we included a qualitative research approach, collecting descriptions of thoughts and emotions from participants through observation and interviews. 

The focus is on exploring subjective experiences, with the aim to better clarify the impact of the intervention on the participants and to uncover insights and meanings of the experience. 

The filmmaking programme represented a growth opportunity not only for the young participants, but also for all the professionals that contributed to the realization of the project. The power of this experience and the impact it can have on individuals is evident in the following reports. All of them shared a common experience of learning new skills, feeling like professionals, and becoming part of a group. 

### 4.1. The Point of View of the Filmmaking Master 

The filmmaking master was highly enthusiastic about the experience of mentoring the students in this pilot study. He commented positively on such issues as the changes he saw in the students, the positive impact on the participants, and how impressed he was with their participation. Figure 1 shows a behind-the-scenes photo shot on the film set.

As the filmmaker noted: 

“For many years I have been carrying out intensive filmmaking courses, including academic ones, in various parts of the world. I have had students of all kinds, preparation, culture, age, language, and professions, but this has genuinely been my most surprising, gratifying and pleasant experience. Implementing a cinema project with such young kids was a challenge. Knowing then that these students had a difficult background and had psychiatric disorders left me uncertain about the possibility of helping them in my role as a director. But, surprisingly, I could see them following the first lessons with interest, travelling together on the bus for site inspections, casting, auditions and becoming more and more passionate. Then I saw them write their stories enthusiastically, get in front of and behind the cameras, stand at the microphones, and take the clapperboard. I saw them smile, communicate, and participate as a real team in an often tiring job full of responsibility and moments of stress. They came out stronger, aware of their abilities and proud to participate in something unique and unrepeatable. This was the real result that gratified me and made me proud. But even the strictly cinematographic results have been beyond all expectations. The stories of these short films, so intense and derived from an authentic and painful experience, have impressive communicative power. I called these young boys and girl “little giants” because of what they were capable of doing. But I would also call them little “seismographs” due to their incredible sensitivity in recording even the slightest painful movements of the world around them”.

### 4.2. The Narration from the Students

The participants of the filmmaking course were also elated to be part of the programme. The course provided a space for these young individuals to explore their creativity and discover their strengths, ultimately leading to a sense of empowerment. Through their words, we see how the magic of filmmaking can transcend barriers and inspire young minds to pursue their dreams.

Federico, a 13-year-old with a diagnosis of OCD and tic disorder says:


*“It is impossible to describe in a few words the magic emotions that I experienced during the course. All of us learned new abilities. We felt like professionals, and we had the responsibility to use complex equipment. But above all, we felt like a big family. Unfortunately, it’s all over, but everything we have done is still within us, in a special place in our heart. After this course I’ve decided that when I grow up, I’ll work in the movie industry”.*


Alessio, a 12-year-old with dyslexia and autism spectrum disorder commented on the filmmaking experience. Even if the text is ungrammatical and full of errors, the meaning is clearly understandable: 


*“We don’t know if this was a sign of destiny or if we are made to do things that giants do, but no one can express the emotions of this project. (In the sense that it is difficult to describe them). This project was beautiful (Arianna’s thread). From this experience, I understood that each of us has skills. Participating in a movie set with all the other kids was an incredible adventure. I hope more similar things are done. Anyone who has yet to experience it cannot understand.”*
(Figure 2. Alessio’s chat screenshot)

## 5. Discussion

This pilot study aimed to explore the benefits of filmmaking therapy for individuals diagnosed with psychiatric or neurodevelopmental disorders in a consecutive sample of children and adolescents. This research is motivated by finding alternative and innovative ways to help children and adolescents with neuropsychiatric disorders in managing life challenges. 

In this work, we measured the benefits of a filmmaking course by applying standardized scales widely used in the clinical assessment of developmental age. A systematic outcome measure was carried out pre- and post-programme in the context of a specialized mental health care team. 

One of the strengths of this study is that it proves the effectiveness of filmmaking projects on children and adolescents affected by various psychiatric disorders, unlike previous studies that had enrolled patients affected by specific disorders such as ASD [11]. The study shows how an inclusive group offers to participants the opportunity to share their different skills and in a highly cooperative environment. It may improve creativity and self-expression, giving the participants a platform to share their perspectives with others. In fact, through storytelling, participants could learn how to convey emotions, communicate ideas, and engage their audience. Moreover, this requires effective communication between team members, and they learn how to explain their ideas clearly and persuasively and how to listen and constructively respond to others.

A second strength of the project Is that It allows the participants to create the stories and, therefore, to express their inner thoughts. The participants had the opportunity to translate their ideas and feelings into cinematic language, one of the most powerful and complete means of expression. They could learn to empathize with their characters and explain their message in a way that resonates with their audience. 

The increase in the number of participants with high self-esteem we have noticed suggests that some of them changed the opinion they have of themselves. Improving the self-esteem or confidence of children and adolescents with psychiatric diseases is one of the main goals of educational interventions, with the purpose to avoid common behaviours, such as avoiding social situations and new and challenging experiences. In accordance with this result, the reduction in withdrawn/depression problems, measured by the Youth Self-Report scale, suggests a positive effect on the internalizing symptoms found in many psychiatric disorders. 

Another important result was the decrease in social problems. Filmmaking requires collaboration and teamwork, as different people with different skills and perspectives come together to create a final product. Moreover, the “Social awareness” and the “Social motivation subscale”, measured by the Social Responsiveness Scale, increased in all participants. Given that some of the patients were diagnosed as having ASD and that the SRS also offered a reasonable interpretation of characteristics regarding the improvement of reciprocal social interaction [22], we can expect a positive outcome of the course on social abilities, even in participants affected with ASD. Our data suggest that, as shown in previous studies focused exclusively on children with ASD [11], film production promotes the development of social skills in participants with diverse psychiatric diseases. In this way, participants learn to communicate effectively, listen actively, and provide constructive feedback. These skills are critical in any social setting and could help them develop empathy, respect, and cooperation. In addition, this could stimulate people to appreciate different perspectives and opinions, which is an essential component of social-emotional intelligence.

Our data also suggest that filmmaking could be a powerful tool for children and adolescents, providing creative expression, communication, collaboration, social awareness, and problem-solving opportunities. In fact, the thought problems and rule-breaking behaviours significantly decreased after the course. 

Filmmaking is a complex and challenging creative endeavour that requires the use of a variety of cognitive skills and executive functions (EF) (motor response inhibition, working memory, sustained attention, response variability, and cognitive switching), which are mental control processes needed to carry out goal-directed behaviours and are fundamental to successful daily functioning across the lifespan [32]. Efs refer to a family of top-down mental processes needed to concentrate and pay attention when acting automatically or relying on instinct or intuition would be ill-advised, insufficient, or impossible [33,34,35]. Therefore, Efs are particularly important in children because they support the complex behaviours necessary for successful social interactions [36]. Deficient executive functioning has been implicated in social skills problems for many clinical populations of children. Moreover, EF impairment is commonly observed in many neurodevelopmental disorders, particularly autism spectrum disorder and attention-deficit/hyperactivity disorder. It has been estimated that 41% to 78% of individuals with ASD exhibit executive dysfunctions [37], and around 89% of children with ADHD were classified as impaired on at least one executive function component [38]. 

Lack of executive functioning was also detected in externalizing disorders (oppositional defiant disorders and conduct disorders) and some mood disorders (major depression, bipolar disorder). However, there are some variations in effect sizes. Apart from the classical “cold” EF, other mechanisms, including the so-called “hot” EF, i.e., motivational dysfunction, delay aversion, sensitivity to reward and punishment, and emotional processing, could be involved.

Creating a film involves many stages, from writing the script to planning the shots, filming, and editing. These stages require using a range of cognitive skills, including all the critical components of executive functioning. 

Planning and organization are fundamental, from writing the script to scheduling the shoot and arranging locations and equipment. Filmmakers need to remember many details, such as camera angles, lighting, and sound while filming. It requires working memory, which is remembering information while using it to complete a task. Additionally, filming requires sustained attention to detail and the ability to switch focus between different aspects of the production. On the other hand, during the production schedule, it could be common to deal with unexpected changes, such as finding creative solutions to technical issues. It could help to improve the executive function of problem solving, which is the ability to identify and solve problems creatively and efficiently. Moreover, making decisions throughout the production process, such as selecting the location, choosing camera angles, and deciding on the best takes to use in the final edit, could enhance the decision-making steps.

During their work, participants had to be flexible and adapt to the changes in the group and the environment, maintain concentration by reducing external interference and distractions, and tolerate denials, frustrations, and routine changes. 

In conclusion, participants could learn how to analyse situations, identify potential problems, and develop creative solutions. Consequently, this suggests that filmmaking, in the same way as other psychosocial interventions, could positively affect some executive functions. In the case of ADHD subjects, this can lead to the reduction in hyperactivity and inattention. This was not recorded in our work, due to the short time of the project and to the lack of psychometric assessments specifically for cognitive functions. We could hypothesize that more prolonged work might improve these outcomes.

Last but not least, thanks to this experience, which requires technology and media tools, participants could learn how to use cameras, edit software, and use other tools to create a final digital product. Some of the participants were inspired by the project to work in the cinema field in the future. This is another important result for adolescents having difficulty in designing their personal, social, and educational development.

The study had some potential limitations, represented by the small size of the sample and the absence of a control group, that limit the generalizability of the findings. In addition, it was conducted with limited time and resources, which can reduce the ability to collect detailed data. However, on the other hand, pre- and post-test comparisons were tested, and different statistically significant results were found. The group encompassed subjects with different and multiple diagnoses, and most participants affected by neurodevelopmental disorders show co-occurring psychiatric symptoms. However, the group was rather homogeneous since the participants shared neuropsychological and affective profiles and developmental trajectories. Additional research is needed in order to explore the effect of this kind of intervention on separate groups. Considering that the sample was not so wide, this was a denotative result, and with a larger sample, more significant results would emerge. More extensive studies will be necessary to test and refine research methods, procedures, and instruments. Larger patient samples and a systematic collection of standardized outcome measures are also needed. As more research is conducted in this area, we will better understand the benefits and limitations of this project as a therapeutic intervention.

## 6. Conclusions

This is a pilot study, but the preliminary results demonstrate that a filmmaking programme can result in measurable changes in emotional, behavioural, and social domains in children and adolescents with psychiatric or neurodevelopmental disorders. The projects that involve movies or videos could help develop new skills, increase awareness, and offer new ways of thinking about specific problems in various patient populations. Projects like this may also encourage children and adolescents with psychiatric disorders to develop skills in their personal and academic lives, explore their passions, and engage with the world around them.

Findings from this research project can offer an empirical basis for the effectiveness of a structured filmmaking method to be replicated in broader contexts (e.g., school and communities) in promoting children’s psychological well-being.

In conclusion, the filmmaking programme proved to be a valid and complete psychosocial intervention to promote mental well-being in general and to strengthen specific weaknesses in young patients suffering from a psychiatric disorder. 

## Figures and Tables

**Figure 1 healthcare-11-01695-f001:**
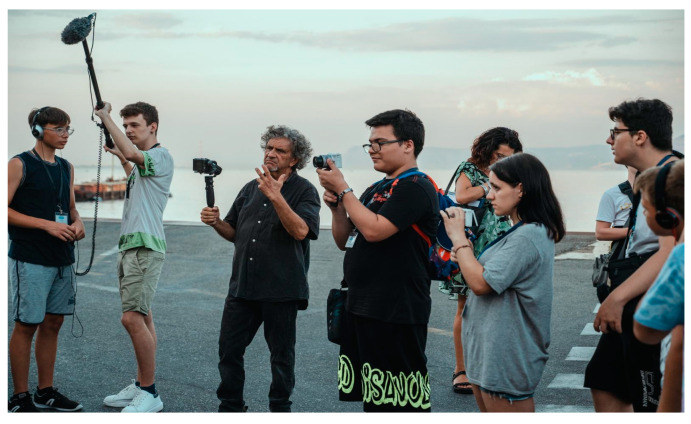
A behind-the-scenes photo shot on the film set.

**Figure 2 healthcare-11-01695-f002:**
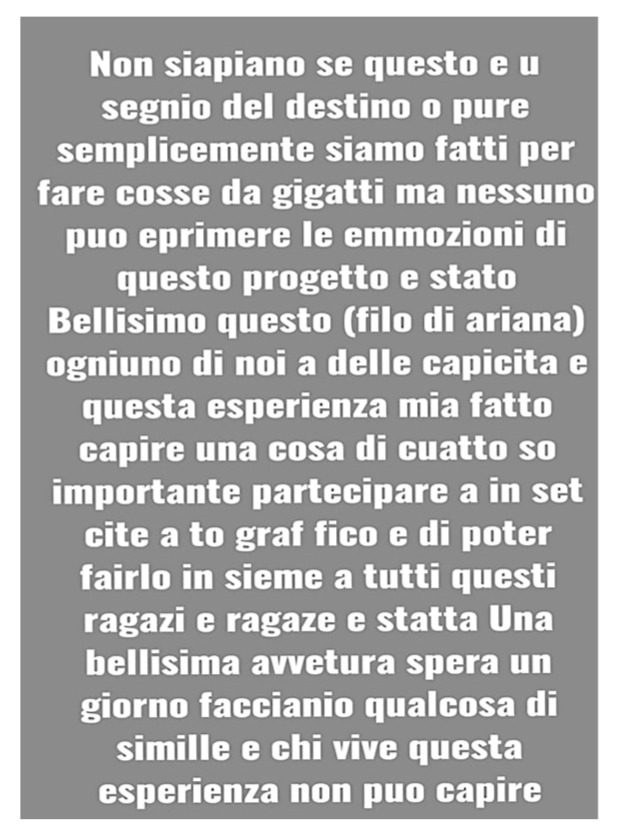
Alessio’s chat screenshot.

**Table 1 healthcare-11-01695-t001:** Descriptive statistics of scores at single items for every scale used. Youth Self-Report scale (YSR), Rosenberg Self Esteem Scale (RSE), Social Responsiveness Scale (SRS), and Revised Conners’ Parent Rating Scale (CPRS-R). T_0_ = scores before treatment; T_1_ = scores after treatment; M = Means values; DS = Standard Deviation; M = Mean; DS = Standard Deviation; Mdn = Median; Min = lowest score; Max = Highest score.

		*N* = 52 at T_0_					*N* = 52 at T_1_				
		M_0_	DS_0_	Min_0_	Max_0_	Mdn_0_	M_1_	DS_1_	Min_1_	Max_1_	Mdn_1_
YSR	Anxious/depressed	64.00	11.892	50	89	61.00	61.63	12.638	50	89	57.50
	Withdrawn/Depressed	63.87	12.361	50	91	62.00	59.83	13.704	50	100	52.00
	Somatic Complaints	57.33	9.699	50	90	52.00	57.79	11.141	50	93	52.00
	Social problems	61.77	9.893	50	87	59.00	58.38	10.642	50	88	54.00
	Thought problems	59.25	8.277	50	78	58.00	56.00	8.287	50	78	51.00
	Attention problems	58.92	11.596	50	96	54.00	57.58	10.541	50	87	52.00
	Rule-breaking behaviour	55.81	8.889	50	81	51.00	54.38	8.117	50	81	50.00
	Aggressive behaviour	56.67	8.340	50	78	52.00	55.00	6.097	50	72	53.00
RSE	Rosenberg Self-Esteem	16.23	7.183	3	30	17.50	17.87	5.347	3	30	17.00
SRS Dimensions	Social awareness	45.15	27.967	0	90	47.00	40.27	23.185	3	72	49.00
	Social cognition	46.92	27.112	4	90	51.00	45.38	25.461	4	83	51.00
	Social communication	53.94	25.430	8	90	57.00	51.75	24.204	3	85	57.00
	Social motivation	54.75	31.153	5	90	63.00	50.85	29.579	1	90	56.00
	Autistic mannerisms	50.75	28.253	1	90	60.00	48.75	28.165	0	86	55.00
	Total score	66.19	13.393	45	90	65.00	63.12	14.438	42	88	63.50
CPRS-R	Oppositional	59.44	17.875	36	88	49.50	57.92	14.650	38	88	54.50
	Cognitive problems	57.25	14.835	41	87	55.00	55.87	17.029	15	88	49.50
	Hyperactivity	57.38	16.408	35	88	52.00	57.10	17.628	35	90	49.00
	ADHD Index	57.40	15.872	32	93	53.00	56.50	17.968	32	99	48.50

**Table 2 healthcare-11-01695-t002:** Differences (*p* < 0.05) between the YSR, SRS, and CPRS-R measures (T scores) at two different time points (T_0_ vs. T_1_). Youth Self-Report scale (YSR), Rosenberg Self Esteem Scale (RSE), Social Responsiveness Scale (SRS), and Revised Conners’ Parent Rating Scale (CPRS-R). T_0_ = scores before treatment; T_1_ = scores after treatment.

		Z	Two-Tailed Significance Test
YSR	Anxious/depressed—T_1_-T_0_	−1.841 ^a^	0.066
	Withdrawn/Depressed—T_1_-T_0_	−2.779 ^a^	0.005
	Somatic Complaints—T_1_-T_0_	−0.083 ^a^	0.934
	Social problems—T_1_-T_0_	−2.693 ^a^	0.007
	Thought problems—T_1_-T_0_	−3.382 ^a^	0.001
	Attention problems—T_1_-T_0_	−1.235 ^a^	0.217
	Rule-breaking behaviour—T_1_-T_0_	−2.035 ^a^	0.042
	Aggressive behaviour—T_1_-T_0_	−1.353 ^a^	0.176
RSE	Rosenberg Self-Esteem—T_1_-T_0_	−1.536 ^b^	0.125
SRS	Social awareness—T_1_-T_0_	−3.475 ^a^	0.001
	Social cognition—T_1_-T_0_	−1.967 ^a^	0.049
	Social communication—T_1_-T_0_	−2.599 ^a^	0.009
	Social motivation—T_1_-T_0_	−2.175 ^a^	0.030
	Autistic mannerisms—T_1_-T_0_	−1.764 ^a^	0.078
	Total score—T_1_-T_0_	−2.496 ^a^	0.013
CPRS-R	Oppositional—T_1_-T_0_	−0.806 ^a^	0.420
	Cognitive problems—T_1_-T_0_	−0.756 ^a^	0.450
	Hyperactivity—T_1_-T_0_	−0.584 ^a^	0.559
	ADHD Index—T_1_-T_0_	−1.842 ^a^	0.065

^a^ Based on positive ratings. ^b^ Based on negative ratings.

## Data Availability

The data presented in this study are available on request from the corresponding author. The data are not publicly available for privacy reasons due to the presence of sensitive information about the participants.

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
