# Peer review of "Effectiveness of an Educational Filmmaking Project in Promoting the Psychological Well-Being of Adolescents with Emotive/Behavioural Problems"

_healthcare, 2023, doi:10.3390/healthcare11121695_

Round 1
Reviewer 1 Report
I believe that the study's findings are very relevant for improving the quality of life of children and adolescents with psychiatric or neurodevelopmental disorders, using Cinematherapy dynamics.
The the article has conditions to be published, however, the authors must clarify the following aspects:
My request for clarification concerns the age of the selected sample and the inclusion criteria that were defined by the authors.
They state in the article that the participants in the project were aged between 9 and 19 years (line 114), but this age group does not meet the second inclusion criterion (line 123).
This criterion requires that the sample to be included in the study must be aged between 11 and 17 years. Thus, in my opinion, there are participants who were included in the study whose age does not meet the second inclusion criterion, and this aspect should be reviewed so that there are no methodological errors.
Author Response
Thanks for having pointing out this error. Indeed, there was a typo error. We involved 52 participants aged between 9 and 17 years. Unfortunately, we got confused because when we enrolled patients, we had some upper than 17 years. However, because we wanted to administer WISC-IV (Wechsler Intelligence Scale for Children), which tests intellectual ability from ages 6 to 16,11 years old, we had to exclude upper 17 years old subjects. The exclusion criteria (line 130) included also a Total IQ (TIQ) <70, assessed by the Wechsler scale (WISC-IV).
Reviewer 2 Report
The stydy is interesting.
The interrelation between quantitative and qualitative analysis could be enhanced in the results and discussion
Author Response
This is a valuable suggestion. We included the following sentence between the two sections:
In addition to the numerical data that we have collected and analyzes by statistical methods, we included a qualitative research approach, collecting descriptions of thoughts and emotions from participants through observation and interviews. The focus is on exploring subjective experiences, with the aim to better clarify the impact of the intervention on the participants and to uncover insights and meanings of the experience.
Reviewer 3 Report
The paper is interesting, athough there are some points that need further consideration.
First of all, the authors should write a Literature review section. Even though there are minor previous works, the authors should write and support how their work contributes to literature in conjuction with other studies.
The Methodology section should be fixed.
1. What are the research questions and hypotheses? The authors should write a new subsection includin Research design, questions.
2. It would helpful if the authors could include a supplementary file with all scales.
In the Results section, please provide the table captions as they are missing.
The English language is good. Please replace the wiht with with (page2, line 81).
Author Response
The paper is interesting, athough there are some points that need further consideration.
First of all, the authors should write a Literature review section. Even though there are minor previous works, the authors should write and support how their work contributes to literature in conjuction with other studies.
Answer: Thanks for the recommendation. The authors cited the few studies reported in Literature, mostly in the paragraph 1. Introduction. Unfortunately, at the moment we have not been able to dedicate an exclusive part to this aspect because of the scarce data in the literature, according to our research.
The Methodology section should be fixed.
1. What are the research questions and hypotheses? The authors should write a new subsection including Research design, questions.
Answer: Thank you for your comment. We agree and included the below section in the section 2. Materials and Methods, as a new paragraph.
2.2 Aims of the study
This study explores if a therapeutic approach based on a six-weeks filmmaking course would help adolescents with emotional/behavioural problems and neurodevelopmental disorders. We hypothesized that an intensive training course aimed to actualize an artistic and creative product, as a film, could lead to improve emotional and social abilities of a group of adolescents. We questioned if the peer interaction and the cooperative work on a film set, may represent a effective tool for promoting adolescents psychological well-being, as reported in most of the scientific literature [14]
2. It would helpful if the authors could include a supplementary file with all scales.
Answer: Thanks for the suggestion. We included a new section as below. You will find in the appendix A all the scales we used.
Appendix A
Supplementary file 1: All scales we used for testing our sample are reported in this supplementary section.
1. Rosenberg Self Esteem Scale (RSE);
2. Youth Self-Report scale (YSR);
3. Social Responsiveness Scale (SRS);
4. Revised Conners' Parent Rating Scale (CPRS-R).
In the Results section, please provide the table captions as they are missing.
Answer: Thanks for the suggestion. The authors changed the text according to the recommendations.
Reviewer 4 Report
Authors investigate the therapeutical effects of cinematherapy on adolescents’ emotional/behavioral problems. By reviewing the literature, they give reasons for the importance of this type of intervention. This review covers relevant literature.
The „Materials and Methods” chapter presents the participants and the process of the therapeutical program. For measuring the effectiveness of the study, they made the participants complete pre and post questionnaire. The main topics of the questionnaire and the applied tools are appropriate. Statistical analyses are proper, results convince us about the effectiveness of the cinematherapy.
Narrative and qualitative reports are interesting, but in the actual form these narratives are only illustrations, and they don’t have any scientific relevance. I suggest omitting this part.
Discussion and conclusion refer to the presented results. I miss some more convincing argumentation or explanation why cinematherapy differs from other forms of art therapies or what is the peculiarity of it.
Strengthen of the study is that it provides convincing evidence of the effectiveness of cinematherapy.
Author Response
Authors investigate the therapeutical effects of cinematherapy on adolescents’ emotional/behavioral problems. By reviewing the literature, they give reasons for the importance of this type of intervention. This review covers relevant literature.
The „Materials and Methods” chapter presents the participants and the process of the therapeutical program. For measuring the effectiveness of the study, they made the participants complete pre and post questionnaire. The main topics of the questionnaire and the applied tools are appropriate. Statistical analyses are proper, results convince us about the effectiveness of the cinematherapy.
Narrative and qualitative reports are interesting, but in the actual form these narratives are only illustrations, and they don’t have any scientific relevance. I suggest omitting this part.
Discussion and conclusion refer to the presented results. I miss some more convincing argumentation or explanation why cinematherapy differs from other forms of art therapies or what is the peculiarity of it.
Strengthen of the study is that it provides convincing evidence of the effectiveness of cinematherapy.
Answer: We would sincerely thank the reviewer for this wise and fervent commentary that prompt the following statements in the introduction section
“Differently from other figurative arts, cinema allows to earn different psychological benefits, such as the identification with the character’s feelings and emotions, the cathartic effect (learning through the character’s experiences), the improvement of social abilities sharing emotions and experiences with other participants. There is clinical evidence that through the use of movies and follow-up sessions, pre-adolescent children whose parents were going through a divorce, improved their capacity to identify and express their feelings and developed coping skills. A monthly, cinema-therapy group sessions, proposed to a group of girls, allowed them to identify and work through anger and conflicts and to marked their feelings about their family and social relationships. Starting from these evidences, we argue that the cooperative work required for making a film is a collective experience which may have a stronger impact on both emotions and thoughts, than the vision of a film made from other people”
Marsick, E. (2010). Cinematherapy with preadolescents experiencing parental divorce: A collective case study. The Arts in Psychotherapy, 37(4), 311–318.
Bierman, J. S., Krieger, A. R., & Leifer, M. (2003). Group cinematherapy as a treatment modality for adolescent girls. Residential Treatment for Children & Youth, 21(1), 1–15
Reviewer 5 Report
This article reports on the pilot test of a filmmaking project designed to promote well being of Italian adolescents with emotional and behavioral problems. The article has a number of strengths. The study is well designed, using internationally recognized and effectively validated measures of constructs such as self esteem, social awareness, and rule breaking behaviors. It adds to the broader literature on the effectiveness of art therapy with adolescents and other populations. The objectives of the study are clearly defined. The study employs a pretest-posttest design, lending additional validity to the conclusions. Statistical methods used were correctly chosen. A final strength of the study is that it incorporated participants with various psychological disorders, unlike some previous studies that focused on participants with a single disorder.
However, several limitations must be addressed. First, the paper needs close editing to correct a number of grammatical and syntax errors. Here are some of the errors, listed by line number:
Lines 23,25,32,33: remove hyphenations
64: Please correct this sentence. It doesn't make sense as written
67: No one sentence paragraphs. There are three one-sentence paragraphs in a row (beginning on lines 67, 70, and 72). Please address this.
81: Typo on this line. "whit" should be corrected to "with."
83: Use consistent terms. The term "objectives" is used in the final sentence on this line. Line 84 uses the term "goal" and line 94 uses the term "objective." The two terms are not interchangeable. The authors should stick with one.
95: Sentence that begins "The participants received a structured teaching..." should read "The participants received structured lessons..."
107-108: The sentence that begins " In addition, the benefits of the filmmaking course have been estimated on the extent...." should read something like this: "In addition, the effects of the filmmaking course have been assessed by the extent of changes in..." An alternative term for "assessed" could be the term "evaluated."
125: the term "..have been collected" should read "...was collected."
138: the term "recruitment" should be replaced with "a meeting" or something similar.
140: the clause "..participated to a meeting.." should be "..participated in a meeting.."
147: The sentence that begins "First, took place the pre-production stage..." should be rewritten something like this: "First was the pre-production stage in which..."
148: The sentence that begins "Hence participants were assigned..." should begin "Participants were assigned..."
155: The last part of the sentence that begins "Additionally, following creative..." should read "Additionally, following the creative..." Also, the final part of the sentence doesn't make sense as written. The part that is of concern reads "...that were chosen by the telling themselves according to the..." This section needs to be rewritten for clarity.
162: "adolescent's" should be written as "adolescents' "
167" "tell" should be "told.."
168: "promote" should be written as "promoted."
173: The sentence that begins "Both to participants..." should read "Both the participants..."
185: The sentence that begins "secondly, they filled..." should read "Second, they completed.."
195: This line needs to be joined with the previous paragraph.
196: The sentence that begins "Last, all primary..." should read "Finally, all primary.."
198: First sentence should begin "The Social Responsiveness Scale.."
208: First sentence should begin "The Revised Conner's Parent..."
156: the phrase "takes place." should read as follows: "took place.."
276: Last word in this line should be "them."
Other errors are present but this should be enough to illustrate the issues.
Line 73 needs at least one citation to document the assertion that is made in this sentence.
Additionally, there are a number of paragraphs that are only one sentence in length. These generally need to be reviewed and combined with the previous or following paragraph.
Readability of Tables 1 and 2 needs to be improved. Readability of both tables would be improved substantially with better formatting and a title for each table. The tables are hard to interpret without a title. The headings in Table 2 such as "Anxious/depressed" would be easier to read if all are left justified. The table would also look neater. This column should also have a title, as do the next two columns. This would promote ease of interpretation. The abbreviations in the rows of Tables 1 and 2 make the tables harder to read. For example DS is not clear. The row titles should be evaluated to promote faster readability.
Some of the statistics reported in the paragraph that begins on line 230 are not in the table. For example, the number of participants with low self esteem at Time 1 and the number at time 2 are not listed Table 1. Simply put, Table 1 needs to have the number of respondents listed for each of the scales for each of the instruments. The N should be listed for each time (Time1 and Time 2).
Section 4 (Narrative and qualitative reports) should have some sort of transition. In its present condition, this section looks like out has just been dropped into the paper. Some sort of transition is needed. Further, the long quote from the filmmaking master needs to be contextualized as presently it just reads like it was dropped in there. Something such as "The filmmaking master was highly enthusiastic about the experience of mentoring the students in this pilot study. He commented positively on such issues as the changes he saw in the students, the positive impact on the participants, and how impressed he was with their participation. As the filmmaker noted:
For many years....."
In other words, some narrative should be provided to lead into the lengthy quote from the filmmaker, such as in my example above. Something similar is needed for the "narration from the students" section. There is only one quote from a student. This is followed by a discussion of Figure 1. Some transition would be very helpful here. Something like "Figure 1 illustrates a chat screenshot in which ...." In other words, the text there is fine. I just recommend a more specific transition such as what I have included.
On the first and second readings, I was unclear if all of the participants had been diagnosed with psychological and neurological disorders or if some had psychological disorders and some had only neurological disorders. After reading again, I noted that the authors state that "most" had more than one diagnosis. However, it is still not clear if or how many had psychological and neurological diagnoses. If some only had neurological disorders, for example, then this must be made clearer in the text and must be taken into account in the presentation of results. In short, if these constitute two separate groups the results should be presented separately for each group and compared. If participants had dual diagnoses of psychological and neurological disorders, additional research is needed that examines adolescent participants who did not have dual diagnoses so that research can better understand the impact of participation in filmmaking on the separate groups (perhaps filmmaking works better for those with psychological disorders than for those with just neurological disorders).
There is a lack of clarity about participant ages. Line 114 says that participants were aged 9 to 19. However, the inclusion criteria on lines 122-123 states the participants had to be between 11 and 17 years old. The authors need to explain why they did not follow their inclusion criteria.
It appears that each of the two "runs" of the project were 6 weeks in length. It's clear that the first phase in each of the two sessions was 3 weeks long. The second three week period is not specifically mentioned, though it appears to have included pre-production, creative, technical, and the filmmaking phase. To make this clear to the reader, line 145 should read as follows:
"In the second phase, which lasted for three weeks....""
This paper reports encouraging and suggestive results from this pilot study. The paper can be significantly improved by attending to these issues above.
This article reports on the pilot test of a filmmaking project designed to promote well being of Italian adolescents with emotional and behavioral problems. The article has a number of strengths. The study is effectively designed, using internationally recognized and effectively validated measures of constructs such as self esteem, social awareness, and rule breaking behaviors. It adds to the broader literature on the effectiveness of art therapy with adolescents and other populations. The objectives of the study are clearly defined. The study employs a pretest-posttest design, lending additional validity to the conclusions. Statistical methods used were correctly chosen. A final strength of the study is that it incorporated participants with various psychological disorders, unlike some previous studies that focused on participants with a single disorder.
However, several limitations must be addressed. First, the paper needs close editing to correct a number of grammatical and syntax errors. Here are some of the errors, listed by line number:
Lines 23,25,32,33: remove hyphenations
64: Please correct this sentence. It doesn't make sense as written
67: No one sentence paragraphs. There are three one-sentence paragraphs in a row (beginning on lines 67, 70, and 72). Please address this.
81: Typo on this line. "whit" should be corrected to "with."
83: Use consistent terms. The term "objectives" is used in the final sentence on this line. Line 84 uses the term "goal" and line 94 uses the term "objective." The two terms are not interchangeable. The authors should stick with one.
95: Sentence that begins "The participants received a structured teaching..." should read "The participants received structured lessons..."
107-108: The sentence that begins " In addition, the benefits of the filmmaking course have been estimated on the extent...." should read something like this: "In addition, the effects of the filmmaking course have been assessed by the extent of changes in..." An alternative term for "assessed" could be the term "evaluated."
125: the term "..have been collected" should read "...was collected."
138: the term "recruitment" should be replaced with "a meeting" or something similar.
140: the clause "..participated to a meeting.." should be "..participated in a meeting.."
147: The sentence that begins "First, took place the pre-production stage..." should be rewritten something like this: "First was the pre-production stage in which..."
148: The sentence that begins "Hence participants were assigned..." should begin "Participants were assigned..."
155: The last part of the sentence that begins "Additionally, following creative..." should read "Additionally, following the creative..." Also, the final part of the sentence doesn't make sense as written. The part that is of concern reads "...that were chosen by the telling themselves according to the..." This section needs to be rewritten for clarity.
162: "adolescent's" should be written as "adolescents' "
167" "tell" should be "told.."
168: "promote" should be written as "promoted."
173: The sentence that begins "Both to participants..." should read "Both the participants..."
185: The sentence that begins "secondly, they filled..." should read "Second, they completed.."
195: This line needs to be joined with the previous paragraph.
196: The sentence that begins "Last, all primary..." should read "Finally, all primary.."
198: First sentence should begin "The Social Responsiveness Scale.."
208: First sentence should begin "The Revised Conner's Parent..."
156: the phrase "takes place." should read as follows: "took place.."
276: Last word in this line should be "them."
Other errors are present but this should be enough to illustrate the issues.
Line 73 needs at least one citation to document the assertion that is made in this sentence.
Readability of Tables 1 and 2 needs to be improved. Readability of both tables would be improved substantially with better formatting and a title for each table. The tables are hard to interpret without a title. The headings in Table 2 such as "Anxious/depressed" would be easier to read if all are left justified. The table would also look neater. This column should also have a title, as do the next two columns. This would promote ease of interpretation. The abbreviations in the rows of Tables 1 and 2 make the tables harder to read. For example DS is not clear. The row titles should be evaluated to promote faster readability.
Some of the statistics reported in the paragraph that begins on line 230 are not in the table. For example, the number of participants with low self esteem at Time 1 and the number at time 2 are not listed Table 1. Simply put, Table 1 needs to have the number of respondents listed for each of the scales for each of the instruments. The N should be listed for each time (Time1 and Time 2).
Section 4 (Narrative and qualitative reports) should have some sort of transition. In its present condition, this section looks like out has just been dropped into the paper. Some sort of transition is needed. Further, the long quote from the filmmaking master needs to be contextualized as presently it just reads like it was dropped in there. Something such as "The filmmaking master was highly enthusiastic about the experience of mentoring the students in this pilot study. He commented positively on such issues as the changes he saw in the students, the positive impact on the participants, and how impressed he was with their participation. As the filmmaker noted:
For many years....."
In other words, some narrative should be provided to lead into the lengthy quote from the filmmaker, such as in my example above. Something similar is needed for the "narration from the students" section. There is only one quote from a student. This is followed by a discussion of Figure 1. Some transition would be very helpful here. Something like "Figure 1 illustrates a chat screenshot in which ...." In other words, the text there is fine. I just recommend a more specific transition such as what I have included.
On the first and second readings, I was unclear if all of the participants had been diagnosed with psychological and neurological disorders or if some had psychological disorders and some had only neurological disorders. After reading again, I noted that the authors state that "most" had more than one diagnosis. However, it is still not clear if or how many had psychological and neurological diagnoses. If some only had neurological disorders, for example, then this must be made clearer in the text and must be taken into account in the presentation of results. In short, if these constitute two separate groups the results should be presented separately for each group and compared. If participants had dual diagnoses of psychological and neurological disorders, additional research is needed that examines adolescent participants who did not have dual diagnoses so that research can better understand the impact of participation in filmmaking on the separate groups (perhaps filmmaking works better for those with psychological disorders than for those with just neurological disorders).
There is a lack of clarity about participant ages. Line 114 says that participants were aged 9 to 19. However, the inclusion criteria on lines 122-123 states the participants had to be between 11 and 17 years old. The authors need to explain why they did not follow their inclusion criteria.
It appears that each of the two "runs" of the project were 6 weeks in length. It's clear that the first phase in each of the two sessions was 3 weeks long. The second three week period is not specifically mentioned, though it appears to have included pre-production, creative, technical, and the filmmaking phase. To make this clear to the reader, line 145 should read as follows:
"In the second phase, which lasted for three weeks....""
This paper reports encouraging and suggestive results from this pilot study. The paper can be significantly improved by attending to these issues above.
Author Response
This article reports on the pilot test of a filmmaking project designed to promote well being of Italian adolescents with emotional and behavioral problems. The article has a number of strengths. The study is well designed, using internationally recognized and effectively validated measures of constructs such as self esteem, social awareness, and rule breaking behaviors. It adds to the broader literature on the effectiveness of art therapy with adolescents and other populations. The objectives of the study are clearly defined. The study employs a pretest-posttest design, lending additional validity to the conclusions. Statistical methods used were correctly chosen. A final strength of the study is that it incorporated participants with various psychological disorders, unlike some previous studies that focused on participants with a single disorder.
However, several limitations must be addressed. First, the paper needs close editing to correct a number of grammatical and syntax errors. Here are some of the errors, listed by line number:
Lines 23,25,32,33: remove hyphenations
Answer: We removed hyphenations of lines 23, 25, 32, 33.
64: Please correct this sentence. It doesn't make sense as written
Answer: We changed the sentence of line 64 in:
Over the past 30 years, a large increase in the number of children and adolescents suffering from anxiety and major depressive disorders or behavioral and conduct disorders has been described. Worldwide, 8.8% of pediatric population have been diagnosed with several mental illnesses, accounting for a heavy disease burden. This represents a challenge for the mental health care system with a growing need of new therapeutic and preventive interventions, such as those based on music, theatre and film therapy”.
67: No one sentence paragraphs. There are three one-sentence paragraphs in a row (beginning on lines 67, 70, and 72). Please address this.
Answer: We modified the one sentence paragraphs of line 67, 10 and 72 in: “Although video-based intervention and filmmaking have been previously studied in adult psychotherapy, their efficacy with children and adolescents still needs to be explored. In fact, in the current literature only few studies on recovery of social skills for children with Autism Spectrum Disorder or intellectual disabilities are available. Studies documenting the usefulness of filmmaking programs, specifically designed for children and adolescents with neurodevelopmental and emotional disorders are very rare.”
81: Typo on this line. "whit" should be corrected to "with."
Answer: The correct sentence is: “This study shows how a film-based intervention, with typical structural and practical phases of the filmmaking process, can actively involve children and adolescents with mental health problems.”
83: Use consistent terms. The term "objectives" is used in the final sentence on this line. Line 84 uses the term "goal" and line 94 uses the term "objective." The two terms are not interchangeable. The authors should stick with one.
Answer: We changed the term “goal” with “objective” in order to be consistent.
95: Sentence that begins "The participants received a structured teaching..." should read "The participants received structured lessons..."
Answer: We changed the sentence in: "The participants received structured lessons..." as you recommended.
107-108: The sentence that begins " In addition, the benefits of the filmmaking course have been estimated on the extent...." should read something like this: "In addition, the effects of the filmmaking course have been assessed by the extent of changes in..." An alternative term for "assessed" could be the term "evaluated."
Answer: We changed the mentioned sentence in "The participants received structured lessons..." as you recommended.
125: the term "..have been collected" should read "...was collected."
Answer: The correct sentence is: “Information regarding personal data (i.e., gender and date of birth), personal history (i.e., previous diagnosis) and familiar history (psychiatric conditions) was collected.”
138: the term "recruitment" should be replaced with "a meeting" or something similar.
Answer: We modified the sentence of line 138 in: “The interested parents were invited for a meeting in order to explain them all details and scope of the project, providing also a written consent for child’s participation.”
140: the clause "..participated to a meeting.." should be "..participated in a meeting.."
Answer: We changed the sentence in: “Subsequently, children participated in a meeting to receive information about the filmmaking project, and to obtain their assent for voluntary involvement.”
147: The sentence that begins "First, took place the pre-production stage..." should be rewritten something like this: "First was the pre-production stage in which..."
Answer: The corrected sentence is: “First of all, took place the pre-production stage in which the participants, divided into groups, ideated and wrote many screenplays.”
148: The sentence that begins "Hence participants were assigned..." should begin "Participants were assigned..."
Answer: The corrected sentence is: “Participants were assigned different filmmaking roles according to their interests and attitudes: scriptwriting, direction, photography, sound, costume design, make-up artistry, production organization, and digital editing.”
155: The last part of the sentence that begins "Additionally, following creative..." should read "Additionally, following the creative..." Also, the final part of the sentence doesn't make sense as written. The part that is of concern reads "...that were chosen by the telling themselves according to the..." This section needs to be rewritten for clarity.
Answer: The correct sentence is: “Additionally, following the creative (i.e., choice of the topic, narratives, roles, etc.) and technical phases (i.e., training on camera functioning), the filmmaking process took place in natural locations or public areas (sports ground, square, etc.), that were chosen by the participants.”
162: "adolescent's" should be written as "adolescents' "
Answer: We changed “adolescent's" in "adolescents' ".
167" "tell" should be "told.."
Answers: We modified “tell” in “told”.
168: "promote" should be written as "promoted."
Answers: We modified “promote” in “promoted”.
173: The sentence that begins "Both to participants..." should read "Both the participants..."
Answers: We modified the sentence in "Both the participants and their caregivers were asked to complete the following self-administered questionnaires at the beginning (T0) and at the end (T1) of the course:”
185: The sentence that begins "secondly, they filled..." should read "Second, they completed.."
Answers: We modified the sentence: “Secondly, they completed the Youth Self-Report Scale, which measures the severity of emotional and behavioral problems based on a sense of self-estimated evaluation.”
195: This line needs to be joined with the previous paragraph.
Answer: We joined the paragraphs.
196: The sentence that begins "Last, all primary..." should read "Finally, all primary.."
Answers: We modified the sentence: “Finally, all primary family caregivers were asked to complete Social Responsiveness Scale (SRS) and the Revised Conners' Parent Rating Scale (CPRS-R) at points T0 and T1.”
198: First sentence should begin "The Social Responsiveness Scale.."
Answer: We modified the sentence: “The Social Responsiveness Scale (SRS) measures the severity of social impairments in children affected by autism spectrum disorder or children from the general population. This questionnaire has excellent psychometric characteristics, and it resulted also a useful tool for verifying changes in the severity of social impairment following intervention [17].”
208: First sentence should begin "The Revised Conner's Parent..."
Answer: We modified the sentence: “The Revised Conners' Parent Rating Scale (CPRS-R; Conners, 1997, Italian adaption Nobile, Alberti & Zuddas, 2007) is a questionnaire measuring parental perception of a child’s behavioral problems. The short form (27 items) comprises four subscales:”
156: the phrase "takes place." should read as follows: "took place.."
Answer: We modified the sentence: “Additionally, following the creative (i.e., choice of the topic, narratives, roles, etc.) and technical phases (i.e., training on camera functioning), the filmmaking process took place in natural locations or public areas (sports ground, square, etc.), that were chosen by the participants.”
276: Last word in this line should be "them."
Answer: We modified the sentence to: “But I would also call them miniature "seismographs" due to their incredible sensitivity in recording even the slightest painful movements of the world around them”.
Other errors are present but this should be enough to illustrate the issues.
Line 73 needs at least one citation to document the assertion that is made in this sentence.
Answer: We adjusted by adding the right citation for the sentence in line 73. “Some film-based interventions have been proposed to adolescents as educational tools, but their efficacy in mental health education remains largely under-explored” (Goodwin J, Saab MM, Dillon CB, Kilty C, McCarthy A, O'Brien M, Philpott LF. The use of film-based interventions in adolescent mental health education: A systematic review. J Psychiatr Res. 2021 May;137:158-172. doi: 10.1016/j.jpsychires.2021.02.055. Epub 2021 Feb 28. PMID: 33677219)
Additionally, there are a number of paragraphs that are only one sentence in length. These generally need to be reviewed and combined with the previous or following paragraph.
Readability of Tables 1 and 2 needs to be improved. Readability of both tables would be improved substantially with better formatting and a title for each table. The tables are hard to interpret without a title. The headings in Table 2 such as "Anxious/depressed" would be easier to read if all are left justified. The table would also look neater. This column should also have a title, as do the next two columns. This would promote ease of interpretation. The abbreviations in the rows of Tables 1 and 2 make the tables harder to read. For example DS is not clear. The row titles should be evaluated to promote faster readability.
Some of the statistics reported in the paragraph that begins on line 230 are not in the table. For example, the number of participants with low self esteem at Time 1 and the number at time 2 are not listed Table 1. Simply put, Table 1 needs to have the number of respondents listed for each of the scales for each of the instruments. The N should be listed for each time (Time1 and Time 2).
Answer: Thanks for pointing it out. We modified Tables 1 and 2 and their titles. We modified Table 2 as you recommended.
Section 4 (Narrative and qualitative reports) should have some sort of transition. In its present condition, this section looks like out has just been dropped into the paper. Some sort of transition is needed. Further, the long quote from the filmmaking master needs to be contextualized as presently it just reads like it was dropped in there. Something such as "The filmmaking master was highly enthusiastic about the experience of mentoring the students in this pilot study. He commented positively on such issues as the changes he saw in the students, the positive impact on the participants, and how impressed he was with their participation. As the filmmaker noted:
For many years....." In other words, some narrative should be provided to lead into the lengthy quote from the filmmaker, such as in my example above. Something similar is needed for the "narration from the students" section. There is only one quote from a student. This is followed by a discussion of Figure 1. Some transition would be very helpful here. Something like "Figure 1 illustrates a chat screenshot in which ...." In other words, the text there is fine. I just recommend a more specific transition such as what I have included.
Answer: Thanks for the commentary. The Authors provide to modify this paragraph in this way below.
4. Narrative and Qualitative Reports
The filmmaking program represented a growth opportunity not only for the young participants, but also for all the professionals that contributed to the realization of the project. The power of this experience and the impact it can have on individuals is evident in the following reports. All of them shared a common experience of learning new skills, feeling like professionals, and becoming part of a group.
4.1. The point of view of the filmmaking master
The filmmaking master was highly enthusiastic about the experience of mentoring the students in this pilot study. He commented positively on such issues as the changes he saw in the students, the positive impact on the participants, and how impressed he was with their participation. As the filmmaker noted:
“For many years I have been carrying out intensive filmmaking courses, including academic ones, in various parts of the world. I have had students of all kinds, preparation, culture, age, language, and professions, but this has genuinely been my life's most surprising, gratifying and pleasant experience. Trying to "make cinema" with such young kids was already challenging. Knowing then that these students came from painful experiences and had psychiatric disorders left me uncertain about the possibility of helping them with my profession. But, surprisingly, I saw them following the first lessons with interest, travelling together on the bus for site inspections, making casts, auditions and becoming more and more passionate. Then I saw them write their stories passionately, get in front of and behind the cameras, stand at the microphones, and take the clapperboard. I saw them smile, communicate, and participate as a real team in an often tiring job full of responsibility and moments of stress. They came out stronger, aware of their abilities and proud to participate in something unique and unrepeatable. This was the real result that gratified me and made me proud. But even the strictly cinematographic results have been beyond all expectations. These little films' stories, so intense and elaborated from an authentic and painful experience, have impressive communicative power. I called these kids "little giants" for what surprising they were capable of doing. But I would also call them miniature "seismographs" due to their incredible sensitivity in recording even the slightest painful movements of the world around them”.
4.2. The narration from the students
The participants of the filmmaking course were also elated to be part of the program. The course provided a space for these young individuals to explore their creativity and discover their strengths, ultimately leading to a sense of empowerment. Through their words, we see how the magic of filmmaking can transcend barriers and inspire young minds to pursue their dreams.
Federico, a 13 years old with a diagnosis of OCD and Tic Disorder says:
“It is impossible to describe in few words the magic emotions that I lived during the course. All of us learned new abilities. We felt like professionals, and we had the responsibility to use complex machines. But above all, we felt a big family. It’s too bad it’s all over, but everything we have done is still inside of us, in a big place inside our heart. After this course I’ve decided that when I grow up, I’ll work in the movie industry”.
Also Alessio, a 12 years old with Dyslexia and Autism Spectrum Disorder commented the filmmaking experience. Even if the text is ungrammatical and full of errors, the meaning is clearly understandable:
"We don't know if this was a sign of destiny or if we are made to do things that giants do, but no one can express the emotions of this project. (In the sense that it is difficult to describe them). This project was beautiful (Arianna's thread). From this experience, I understood that each of us has skills. Participating in a movie set with all the other kids was an incredible adventure. I hope more similar things are done. Anyone who has yet to experience it cannot understand.” (Figure 1. Alessio's chat screenshot)
On the first and second readings, I was unclear if all of the participants had been diagnosed with psychological and neurological disorders or if some had psychological disorders and some had only neurological disorders. After reading again, I noted that the authors state that "most" had more than one diagnosis. However, it is still not clear if or how many had psychological and neurological diagnoses. If some only had neurological disorders, for example, then this must be made clearer in the text and must be taken into account in the presentation of results. In short, if these constitute two separate groups the results should be presented separately for each group and compared. If participants had dual diagnoses of psychological and neurological disorders, additional research is needed that examines adolescent participants who did not have dual diagnoses so that research can better understand the impact of participation in filmmaking on the separate groups (perhaps filmmaking works better for those with psychological disorders than for those with just neurological disorders).
Answer: All participants had received a psychiatric diagnosis (such as anxiety disorder or mood disorder) and/or a diagnosis of neurodevelopmental disorder (ADHD, ASD, Motor disorders, specific learning disabilities). None of them had “neurological disorders (such as cerebral palsy or epilepsy). As commonly observed, most of participants affected with neurodevelopmental disorders show co-occurring psychiatric symptoms. Even though the group encompassed subjects with different diagnosis, it was rather homogeneous since the participants shared neuropsychological and affective profiles and developmental trajectories.
We agree with the reviewer that additional research is needed in order to explore the effect of this kind of interventions on separate groups. Nevertheless, the construction of “pure” groups with the same diagnosis or very similar profiles, is a very hard issue in the clinical practice. It would be easier to enroll more homogenous samples in specific programs or residential service for adolescents.
We have added the following statements among limits
The group encompassed subjects with different and multiple diagnosis and most of participants affected with neurodevelopmental disorders show co-occurring psychiatric symptoms. However, the group was rather homogeneous since the participants shared neuropsychological and affective profiles and developmental trajectories. Additional research is needed in order to explore the effect of this kind of interventions on separate groups.
There is a lack of clarity about participant ages. Line 114 says that participants were aged 9 to 19. However, the inclusion criteria on lines 122-123 states the participants had to be between 11 and 17 years old. The authors need to explain why they did not follow their inclusion criteria.
Answer: There was a typo error. Our group of patients was aged between 9 and 17 years old, because the inclusion criteria were to be aged under 17 years old. Our research design in fact, planned to use WISC-IV
scale (Wechsler Intelligence Scale for Children) to detect the Total IQ, which represented our exclusion criteria if under 70.
It appears that each of the two "runs" of the project were 6 weeks in length. It's clear that the first phase in each of the two sessions was 3 weeks long. The second three week period is not specifically mentioned, though it appears to have included pre-production, creative, technical, and the filmmaking phase. To make this clear to the reader, line 145 should read as follows:
"In the second phase, which lasted for three weeks....""
Answer: We modified the manuscript from line 142 to line 171 in order to explain more clearly to the reader the procedure and phases of the project:
“Then, participants started a daily course of 5-6 hours per day, lasting in total three weeks. During the first two days, all project members attended theory-based lessons in filmmaking history and techniques. This phase also focused on building trust relationships among the participants. From the third day onward, the three main stages of the film production process were structured. First of all, took place the pre-production stage in which the participants, divided into groups, ideated and wrote many screenplays. Hence participants were assigned different filmmaking roles according to their interests and attitudes: scriptwriting, direction, photography, sound, costume design, make-up artistry, production organization, and digital editing. Some subjects also held more than one role. During the pre-production stage also took place the storyboarding, during which spaces were selected and equipment for sets was provided. A day-by-day film production timeline was also created. Additionally, following the creative (i.e., choice of the topic, narratives, roles, etc.) and technical phases (i.e., training on camera functioning), the filmmaking process took place in natural locations or public areas (sports ground, square, etc.), that were chosen by the participants. Participants were expected to produce a few short films (lasting about 15 minutes), but they outperformed expectations and ultimately produced 13 short films in total (6 in the first edition and 7 in the second edition of the course). The script themes were largely inspired by the main adolescent’s issues and concerns, such as unintentional injuries, violence, mental health, suicide, self-cutting, alcohol and drug use, gender dysphoria, sense of unease, etc. In addition, some scripts were inspired by social and political issues such as the Russia-Ukraine war and the fear of the apocalypse. Only a few themes contain hopeful messages that aim to promote the power of friendship and love. However, every script and film tell a good story with an exciting plot that promotes reflection on sensitive topics. Later, the video-editing crew, with the movie director, worked for the post-production phase: firstly, they drafted a rough cut of the film, then they began to review and edit the footage making additions like visual effects, background music and sound design.”
This paper reports encouraging and suggestive results from this pilot study. The paper can be significantly improved by attending to these issues above.
Reviewer 6 Report
I think this is a very interesting idea for a study. I think several improvements are needed before it would be ready for publication in healthcare.
1) As currently constructed, the theoretical framework does not support logical connections between participating in a filmmaking project and psychological health and well-being among adolescents. The authors are encouraged to provide stronger empirical connections (through supporting literature) and a possible exploration around some of the underlying psychological connections between these factors.
2) The study design, with inclusion of both quantitative and qualitative data, needs to be more clearly articulated. This should include a discussion of whether a mixed-methods approach is desired or being used, and how quantitative and qualitative data inform each other under this design.
3) On page 2, lines 49-50, this statistic needs a citation.
Could use some additional editing for grammar, language and structure.
Author Response
I think this is a very interesting idea for a study. I think several improvements are needed before it would be ready for publication in healthcare.
1) As currently constructed, the theoretical framework does not support logical connections between participating in a filmmaking project and psychological health and well-being among adolescents. The authors are encouraged to provide stronger empirical connections (through supporting literature) and a possible exploration around some of the underlying psychological connections between these factors.
Answer: Thanks for the suggestion. The authors changed the text according to the recommendations, including a new paragraph in the section 2. Materials and Methods, in wich we explain that, the connections between participating in a filmmaking project and psychological health and well-being among troubled adolescents, was right our hypothesis. For example, we wondered if, also in this case, as in most of the scientific literature, the use for examples of structured peer interaction as a therapeutic tool, would have improved the psychophysical state.
2) The study design, with inclusion of both quantitative and qualitative data, needs to be more clearly articulated. This should include a discussion of whether a mixed-methods approach is desired or being used, and how quantitative and qualitative data inform each other under this design.
Yes, we agree. Thus we have added the following sentence in order to introduce the qualitative data:
In addition to the numerical data that we have collected and analyzes by statistical methods, we included a qualitative research approach, collecting descriptions of thoughts and emotions from participants through observation and interviews. The focus is on exploring subjective experiences, with the aim to better clarify the impact of the intervention on the participants and to uncover insights and meanings of the experience.
3) On page 2, lines 49-50, this statistic needs a citation.
Answer: Thanks for the suggestion. There’s a typo on page 2, lines 49-50, we added citations from which we had taken the information.
[2] V. Patel, A. J. Flisher, S. Hetrick, and P. McGorry, “Mental health of young people: a global public-health challenge,” Lancet, vol. 369, no. 9569, pp. 1302–1313, Apr. 2007, doi: 10.1016/S0140-6736(07)60368-7.
[3]U. Panchal et al., “The impact of COVID-19 lockdown on child and adolescent mental health: systematic review,” Eur Child Adolesc Psychiatry, vol. 1, pp. 1–27, Aug. 2021, doi: 10.1007/S00787-021-01856-W/FIGURES/2.
Comments on the Quality of English Language
Could use some additional editing for grammar, language and structure.
We are very sorry for the English mistakes. As suggested, a native English speaker revised the manuscript.
Round 2
Reviewer 3 Report
This version is better than previous one. I would suggest the authors to extend their work by utilising new techniques.
Author Response
Thank you for the review work; we certainly will take up the valuable suggestion.
Reviewer 6 Report
Thank you for addressing my suggested comments. The manuscript is much stronger in its current form.
Author Response
Thank you for the review work.